# Histological and Cytological Characterization of Anther and Appendage Development in Asian Lotus (*Nelumbo nucifera* Gaertn.)

**DOI:** 10.3390/ijms20051015

**Published:** 2019-02-26

**Authors:** Dasheng Zhang, Qing Chen, Qingqing Liu, Fengluan Liu, Lijie Cui, Wen Shao, Shaohua Wu, Jie Xu, Daike Tian

**Affiliations:** 1Shanghai Chenshan Plant Science Research Center of Chinese Academy of Sciences, Shanghai Chenshan Botanical Garden, Shanghai 201602, China; zhdash001@163.com (D.Z.); cancy24@gmail.com (Q.L.); 19liu19@163.com (F.L.); shaowen19792005@163.com (W.S.); 2Shanghai Key Laboratory of Plant Functional Genomics and Resources, Shanghai 201602, China; 3College of Horticulture, Fujian Agriculture and Forestry University, Fuzhou 350002, China; qingchen0523@163.com (Q.C.); wsh6677@hotmail.com (S.W.); 4Development Center of Plant Germplasm Resources, College of Life Science, Shanghai Normal University, Shanghai 200234, China; cuilj@shnu.edu.cn; 5School of Life Sciences and Biotechnology, Shanghai Jiao Tong University, Shanghai 200240, China

**Keywords:** Asian lotus, anther, pollen, microspore, anther appendage

## Abstract

The lotus (*Nelumbo* Adans.) is a perennial aquatic plant with important value in horticulture, medicine, food, religion, and culture. It is rich in germplasm and more than 2000 cultivars have been cultivated through hybridization and natural selection. Microsporogenesis and male gametogenesis in the anther are important for hybridization in flowering plants. However, little is known about the cytological events, especially related to the stamen, during the reproduction of the lotus. To better understand the mechanism controlling the male reproductive development of the lotus, we investigated the flower structure of the Asian lotus (*N. nucifera*). The cytological analysis of anther morphogenesis showed both the common and specialized cytological events as well as the formation of mature pollen grains via meiosis and mitosis during lotus anther development. Intriguingly, an anatomical difference in anther appendage structures was observed between the Asian lotus and the American lotus (*N. lutea*). To facilitate future study on lotus male reproduction, we categorized pollen development into 11 stages according to the characterized cytological events. This discovery expands our knowledge on the pollen and appendage development of the lotus as well as improving the understanding of the species differentiation of *N. nucifera* and *N. lutea*.

## 1. Introduction

The Lotus (*Nelumbo*) is a perennial aquatic plant with ornamental, medicinal, edible and cultural importance [1,2,3]. It has only two living species: the American lotus (*N. lutea* Willd.) and the Asian lotus (*N. nucifera* Gaertn.) [4]. The American lotus has light-yellow colored flowers only and is distributed in North America and the northern regions of South America, while the Asian lotus has a richer diversity in morphology at the population level with wider distribution ranges (Asia, north Oceania and south Russia). The Asian lotus, also called the sacred lotus, has been domesticated in Asia for about 7000 years and is cultivated as a major crop where its rhizomes and seeds are used as vegetables particularly in China and Southeast Asian countries [5]. It is the national flower of India and one of the traditional flowers of China and Vietnam. Furthermore, the Asian lotus plays a significant role in religious and cultural activities in many Asian countries [6]. A long history of artificial hybridization and selection has generated more than 2000 lotus cultivars that differ in plant size, flower color, flower form, flower shape and tepal number [7]. As a famous ornamental plant, lotus is widely planted in water gardens, ponds, lakes and rivers to beautify and purify the environment. It also produces a series of secondary metabolites with important medicinal functions, including flavonoids, alkaloids, triterpenoids, steroids, glycosides and polyphenols from the leaves, flowers, seedpods and seeds [8]. 

The lotus is a long-day plant. Its flowers open in the early morning and the life span of one single flower usually does not exceed four days. The lotus flower is solitary and hermaphrodite, and is composed of the perianth, androecium, gyneceum, receptacle, and peduncle (Figure 1a). The calyx is the outermost whorl of the perianth, usually consisting of four sepals that are green, thick, tenacious, and similar to tetals in structure, except that they fall earlier. One flower contains twenty or more tepals that vary widely in size, shape, number and color from population to population and from cultivar to cultivar, which results in a diverse floral morphology. Usually, a lotus flower is self- incompatible because the multiple stigmas of a receptacle mature ahead of the stamens in the same flower [6]. Up to now, the majority of lotuses have stamens with a few exceptions such as ‘Guangyue Lou’, ‘Miracle’, etc. and the three thousand-petalled-type Asian lotus cultivars ‘Qian Ban’, ‘Yiliang Qianban’, and ‘Zhizun Qianban’ where the stamens are fully transformed into tepals. 

The pollen development of the American lotus has been described and can be divided into five different stages [9]. However, the development of the pollen and anther of Asian lotus, a more economically important species, still remains unknown. Due to a long history of geographical isolation and evolution, large differences exist in the morphology between these two species, especially in the stamen, anther pigment and appendage shape. Although both the American lotus and Asian lotus can easily hybridize with each other to produce fertile seeds, the reproductive ability of their hybrid offspring has largely declined [4,10]. To better understand the process and characteristics of the male reproductive development of the Asian lotus in comparison with previous reports on the American lotus, we investigated the flower and anther structure, anther morphogenesis, and pollen formation of a wild type Asian lotus. Exploring the developmental events in the lotus plant, particularly into its reproductive mechanism, will improve our knowledge on species identity and differentiation.

## 2. Results

### 2.1. The Morphologic Characteristics of the Anther and Its Appendage in N. nucifera

In the underwater flower bud, the stamens originated from the stamen primordium, which came from the lateral edges of the growth cone inside the tepal primordium (Figure 1b) and appeared around the base of the receptacle. Shortly after the formation of the stamen primordium, the pistil primordium formed at the top central domain of the growth cone. Meanwhile, the receptacle gradually developed and elevated incrementally (Figure 1c). Before the anther sac was initiated, the appendage could obviously be observed at the top of the stamen (Figure 1d). Along with stamen development, the anther and appendage gradually elongated and reached 2.5–4.5 cm at full length (Figure 1e,f). At the late stage of anther development, the filament continued to extend while the anther and appendage no longer elongated (Figure 1f). Each stamen contained a filament and an anther with four pollen sacs linked to the filament by connective tissues (Figure 1g). The anther wall consisted of the epidermis, endothecium, middle layer, and tapetum. The sporogenic cells were enclosed in a fluid-filled locule surrounded by tapetal cells (Figure 1g). When the flower was fully open, the fiber of the wall layers became thick and dehiscent, before the mature pollen grains were dispersed.

The appendage of the Asian lotus stamen was long, oval shaped, and lay at the top of the connective (Figure 1f). It is usually milky white or sometimes pink to rosy red in color for Asian lotus cultivars, while the appendage of the American lotus is bright yellow with a sickle-like shape. The appendage only had two layers: the epidermis and hypodermis parenchyma (Figure 1h). The subepidermal parenchyma also contained a high density of granules that could be deeply stained with toluidine blue (Figure 1h). 

### 2.2. The Development of Pollen in N. nucifera

Based on the morphology of pollen development, we divided the pollen development of the lotus into 11 stages, from the differentiation of archesporial cells to mature pollen grain production (Figure 2, Table 1). In general, the typical characteristics of pollen development in the Asian lotus included the microspore mother cells undergoing successive meiosis resulting in the formation of tetrads; separated young microspores, where the nucleus divided mitotically, two celled microspores with generative and vegetative cells, and mature 3-aperture pollen grains. Like *Arabidopsis*, the type of tapetum appeared to be a secretory-type that produced a granule structure (called orbicules), which was assumed to transport tapetum-produced sporopollenin precursors through the hydrophilic cell membranes to the locule during pollen exine development.

The first stage is the sporogenous cell stage. At this stage, the sporogenous cells tightly abutted the polygonal shape and completely filled the locular space (Figure 2a). The cell cytoplasm was stained densely with chromatic stains, and the nucleolus was relatively large and round. Fewer mitochondria and tiny vacuoles could be seen in the cytoplasm (Figure 3a). 

Then, the sporogenous cells generated microspore mother cells within the locule at the microspore mother cell stage (Figure 2b). At this stage, the microspore mother cells were completely separated with a thickened callose wall and the nucleolus could clearly be seen. The tapetal cells were multinucleate and could be observed in the mitosis process (Figure 2b, as indicated by the red arrow). The epidermis, endothecium and middle layer cells contained large vacuoles in the cytoplasm at this stage and cutin (or wax) began to accumulate outside the anther wall (Figure 3c). From the dyads stage to the tetrad stage (stage 3 to 4), the dyads and tetrads formed in sequence through MMC meiosis (Figure 2c,d). From the early young microspore stage (stage 5), the callose wall around the tetrads degraded and free microspores were released (Figure 2e). The primexine developed on the surface of the young microspores (Figure 3d). Remarkably, both the tapetal cells and microspores contained numerous mitochondria and rough endoplasmic reticulum (ER) with expanded cisternae (Figure 3e,f). 

At the middle free spore stage, the typical pollen wall of young microspores was gradually established (Figure 2f, as indicated by the red arrow, stage 6). At the late young microspore stage (stage 7), the microspores became vacuolated and turned into a round shape. The nucleus was pushed by large central vacuoles to one side and the pollen wall became thicker (Figure 2g). Simultaneously, the cytoplasm of the tapetal cells was continuously concentrated with a high electron-dense cytoplasm. Similar to *Arabidopsis*, it appeared to be a secretory-type tapetum that produced a granule structure (called orbicules), which was assumed to transport tapetum-produced sporopollenin precursors through the hydrophilic cell membranes to the locule during pollen exine development (Figure 4a). Along with the formation of the pollen wall, the germination aperture was also initiated from the early young microspore stage, and was obviously visible at the middle spore stage (Figure 4b,c, indicated by white arrow).

At the early bicellular pollen stage (stage 8), the nucleus underwent mitosis with asymmetric cell division to generate vegetative cells and generative cells (Figure 2h, as indicated by the red arrow). The pollen wall became thicker and the tapetum degraded into the hill-like shape. At the late bicellular pollen stage (stage 9), the large central vacuole turned into multiple tiny vacuoles and the tapetum degraded into a strip-like form (Figure 2i). The majority of the mature pollen grains was full of starch grains and had a uniformly dense reticulate ornamentation on the pollen surface at the mature pollen stage (Figure 2j and Figure 4d). Then the mature pollens were released along with the dehiscence of the locule at the anther dehiscence stage (stage 11). The tapetum disappeared and only one middle layer remained (Figure 2k).

### 2.3. The Development of the Pollen Wall in N. nucifera

After male meiotic cytokinesis in the lotus anther, the pollen wall was originally initiated around the surface of individual microspores of the tetrad. The degradative callose wall was the first of several layers deposited on the microspore surface (Figure 5b). As the young microspores were released from the tetrads, primexine (PE) was deposited between the callose wall and plasma membrane (Figure 5d). Besides this, the initial, electron-dense procolumellae (PC) also formed. The nexine II (also called endexine) lamellae was first initiated between the primexine and undulation of the plasma membrane (Figure 5d, as indicated by the white arrow).

With the thickening of the primexine at the early young microspore stage (Figure 5f), fibrillar-like materials (or loose reticulum) started to develop between the primexine and plasma membrane to form the nexine II (Figure 5h). The primexine continued to thicken up to the middle young microspore stage. Then more and more sporopollenin continuously accumulated on the surface of the young microspores to form the columella at the late young microspore stage (Figure 5j). Meanwhile, a white-line-centered nexine I (foot layer) first appeared between sexine and nexine II (Figure 5j,l, white arrow indicates the white-line-center). At the early bicellular pollen stage, the tectum and columella were obviously observed with the gradual thickening of the sporopollenin (Figure 5l). 

The intine of the pollen wall, which is pecto-cellulosic in nature, finally appeared on the periphery of the pollen grain cytoplasm at the late bicellular pollen stage. The sporopollenin further accumulated outside the pollen wall, which resulted in an irregular form of pollen coat. The diameter of the exine thickened further while the diameter of the nexine gradually narrowed at this stage (Figure 5n). When the pollen grains completely matured with a round shape, the pollen wall was characterized by a well-developed intine and a compressed layer of exine. However, the inner nexine was compressed further and was difficult to distinguish from the outer sexine (Figure 5p). Finally, the mature pollen wall contained a well-developed sexine (about 2 μm), a thin layer of nexine (about 0.5 μm), and a thickened layer of intine (about 2 μm). The sexine consisted of a tectum (about 0.5 μm) linked by a pollen coat and irregular columella (about 0.1–0.5 μm) (Figure 5q).

### 2.4. The Development and Structure of the Anther Appendage in the Lotus

From the semi-section observation, the structure of the appendage only had two layers, the epidermis and hypodermis parenchyma (Figure 1h). The inner parenchyma contained numerous mitochondria, vacuoles and starch granules (Figure 6k). The parenchyma cells under the epidermal layer were also filled with spherical osmiophilic bodies that could be densely stained in TEM observation (Figure 6l, as indicated by the red arrow). These spherical bodies accumulated gradually under the epidermis layer along with the development of the appendage (Figure 6e–j). Interestingly, the morphology and structure of the appendage were different between the Asian lotus and the American lotus (Figure 6b and Figure 7d–k). The spherical bodies accumulated with multiple layers under the epidermis cells in the Asian lotus (Figure 7b, as indicated by the red arrow), whereas it only had one layer under the epidermis cells and were randomly scattered in the American lotus at different developmental stages (Figure 7d–j, as indicated by the red arrow). From the SEM observations, the surface appendage was smooth in the Asian lotus whereas it was linearly rough in the American lotus (Figure 6b and Figure 7k). 

## 3. Discussion

### 3.1. The Characteristics of Anther Wall Morphogenesis in N. nucifera

Like other angiosperms, the anther wall of the lotus is composed of four layers of cells: the epidermis, endothecium, middle layer, and tapetum. Pollen maturation relies heavily on the surrounding tissues of the anther wall, especially the tapetum and the middle layer [11]. The innermost tapetum directly wraps the microspore mother cells or the late microspores. The tapetal cytoplasm is rich in mitochondria, endoplasmic reticulum, and Golgi organelles with a strong metabolism. Similar to the American lotus, the tapetum of the Asian lotus is also of the secretory type. Previously, the lotus had an anatomical resemblance to monocots and might possibly represent an evolutionary forerunner of the monocots. It is currently placed along with early diverging eudicots of the Proteales based on phylogenetic analysis [12]. An important feature of monocot pollen is that it has a unique structure, known as an Ubisch body, on the tapetal membrane [13]. However, we did not observe the presence of Ubisch bodies on the lotus tapetal membrane from the TEM observations, which further confirmed that lotus really belongs to eudicots.

The middle layer also has secretory activity and was predicted to play a key role in stamen development and pollen maturation [14,15]. Unlike model plants such as *Arabidopsis* and rice, the important characteristic of the lotus is the multilayered (three or more) middle layer. With the exception of a few plant species, such as the lily, both the tapetum and the middle layer of the major eudicots show secretory activity and undergo programmed cell death (PCD). Usually, the middle layer degenerates before the tapetum in the PCD process [15]. However, in the lotus, the middle layer existed right up to the mature pollen stage and eventually degenerated at the anther dehiscence stage. The precise role of the multiple layers of the middle layer remains unknown and the function of the middle layer is worthy of further study.

### 3.2. The Characteristics of Pollen and Pollen Wall Morphogenesis in N. nucifera

From the semi-section observations, the development of lotus pollen appears to be a well-conserved process in angiosperms, which also experiences the process of reproductive cell division, differentiation to pollen formation, maturation and anther dehiscence, suggesting a conserved mechanism of male reproduction in higher plants. In this process, the nucleus undergoes mitosis with asymmetric cell division to generate vegetative cells and generative cells, which has not been previously reported in the pollen development of the American lotus. However, more evidence should be provided to confirm the existence of this stage in the American lotus. 

From the TEM observation, the development of the pollen wall in the Asian lotus was similar to that of the American lotus except for the length of the anther. In both cases, primexine was initiated at the late tetrad stage and eventually developed into an intact pollen wall comprising exine and intine with the deposition of sporopollenin. Primexine is thought to be a ‘‘template’’ for exine patterning and provides an efficient substructure for sporopollenin deposition, although its formation is still unclear [16]. In the American lotus, a granule layer underlying the endexine was suggested to be an intine precursor [6]. However, from the continuous developmental structure of the pollen wall, this granule layer should be nexine II. 

### 3.3. The Developmental Differences in the Anther and Appendage between N. nucifera and N. lutea

Aside from the differences in flower shape, flower color and tepal number, anther development was also different between the American lotus and the Asian lotus. During anther development in the American lotus, microspore development was later than that of the Asian lotus. Sporogenous cells began to form until the length of the anther reached 2.5–7 mm [9]. However, by the late young microspore stage, the anther lengths of the American lotus and Asian lotus were similar (16–17 mm). After that, the anther length of the Asian lotus no longer lengthened, while the anther length of the American lotus continued to extend to 34–38 mm at the mature stage [9]. 

The appendage lies at the top of the stamen with a long oval-shape, however, the structure and biological function of the appendage are still unclear. The surface structure of the appendage was similar to that of the filament rather than the pollen sac, which indicated that the appendage should be a specialized part of the filaments. The anther appendage in some other plant taxa, such as Ericaceae, *Globba* (Zingiberaceae), *Incarvillea* (Bignoniaceae), *Axinaea* (Melastomataceae), ginger (Zingiberaceae) and violet (*Viola lutea*) plays diverse roles in anther dehiscence and pollen dispersal [17,18,19,20,21,22]. Perching at the top of the receptacle, the anther appendage of the lotus was hypothesized to cool the temperature of the stamens. In addition, it has been proposed to be the source of fragrance in the lotus (for example, in Vietnam, the anther appendages are mixed with tea to increase the fragrance). Intriguingly, the spherical osmiophilic bodies accumulated under the epidermis layer in the Asian lotus, whereas they were randomly scattered in the American lotus. The appendage structure at the different developmental stages also indicated that only one layer of osmiophilic bodies was observed under the epidermis cells in the American lotus. The composition of these spherical bodies and their biological function needs to be clarified in further studies.

## 4. Materials and Methods

### 4.1. Preparation of Plant Material

The Asian lotus ‘Honghu Hong’ (large plant, red and single flower, a wild type introduced from the natural population of Hong Lake in Hubei Province, China) and the American yellow lotus (large plant, light-yellow and single flower, wild type from Miami, FL, USA) were used as experimental materials. Both lotuses were cultivated in the International Nelumbo Collection (INC) located at Shanghai Chenshan Botanical Garden, Shanghai, China. Three floral buds of each lotus were collected at different stages and thereafter ten fresh anthers in each bud were isolated for fixing in FAA solution (formalin-acetic acid alcohol with 50% ethanol, 5% glacial acetic acid and 3.7% formaldehyde). The length of the anthers and appendages were randomly measured from 10 stamens at each sampling time.

### 4.2. Semi-thin Section for Morphological Observation

The fixed sampled anthers with appendages in FAA solution were dehydrated in graded ethanol (70, 80 and 95%) for 30 min at each step, and then dehydrated in 100% ethanol twice for 2 h each time. Subsequently, the samples were embedded in Kulzer liquid Technovit 7100 cold polymerizing resin (Heraeus Kulzer, Basingstoke, UK) at 45 °C [23]. The embedded samples were cut into slices with a thickness of 3 μm with a Leica EM UC7 ultramicrotome (Leica, Wetzlar, Germany). Then the sections were stained with 0.25% Toluidine blue O (Sangon Biotech, Shanghai, China) at 42 °C. The anther sections were observed by an Olympus microscope BX43 and Olympus DP73 digital camera.

### 4.3. Section Observation by Scanning Electron Microscopy

The scanning electron microscopy (SEM) examination was modified from [24]. Briefly, the anthers and appendages at the mature stage were collected, fixed in FAA solution, and dehydrated in graded ethanol (70%, 80%, 95%, and 100%) for each step for 3 min, respectively. Then the dehydrated samples were dried at the critical point temperature (Leica EM CPD300) and coated with 5-nm thickness Aurum with a Leica EM SCD050 ion sputter. Aurum-coated samples were observed under a FEI Quanta 250 scanning electron microscope.

### 4.4. Section Observation by Transmission Electron Microscopy

For the transmission electron microscopy (TEM) analyses, the anthers at different developmental stages were collected and fixed in 2.5% glutaraldehyde solution in 0.1 M sodium phosphate buffer (PBS, pH 7.2). Then the samples were rinsed three times with 0.1 M PBS buffer for 15 min each time and fixed in 2% osmium tetroxide (OsO_4_) in the same buffer [24]. The samples were dehydrated in graded ethanol (50%, 70%, and 90%), 90% acetone, and 100% acetone for 20 min each step, and then embedded in Epon 812 resins [25]. The samples were sectioned to a thickness of 70 nm with a Leica EM UC7 ultramicrotome and then double-stained with 2% (*w*/*v*) uranyl acetate and 2.6% (*w*/*v*) lead citrate aqueous solution. The ultrathin sections were observed under a transmission electron microscope (Tecnai G_2_ Spirit Biotwin, FEI, Hillsboro, OR, USA).

## 5. Conclusions

This is the first report of the pollen and anther development in the Asian lotus. The results expanded our knowledge on the pollen and anther appendage development of *Nelumbo* and will also improve understanding of the identity and differentiation of the two lotus species. The key cytological events of eleven stages of anther and pollen development of the Asian lotus documented in details in this study are a good base to analyze the differences between the lotus species, selecting taxonomically important traits. Description of regular anther and pollen development of the Asian lotus allows recognition of the abnormalities at different stages which could lead to male sterility. Cytological data of anther and pollen development are useful in further crossbreeding experiments used for the commercial production of hybrids between the two lotus species. The comparative analysis of the anther and microspore formation process will provide an important contribution to the evolution of terrestrial and aquatic plants. The anther’s appendage traits discriminate Asian and American lotuses. Determination of the biological function of the anther appendage is an interesting and challenging topic for future study. 

## Figures and Tables

**Figure 1 ijms-20-01015-f001:**
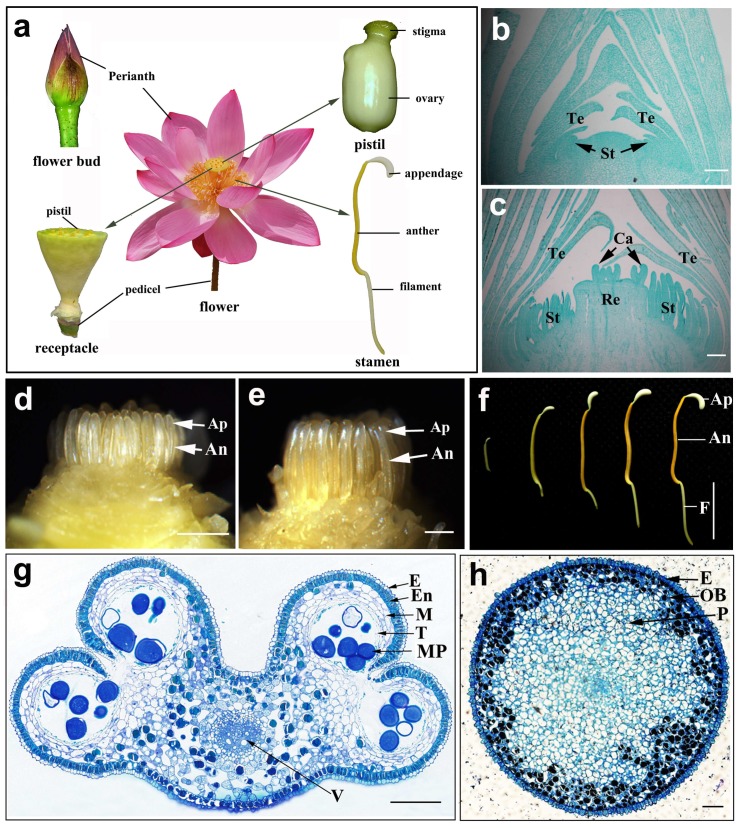
Development and transverse sections of the anther and appendage of *N. nucifera* ‘Honghu Hong’. (**a**) Scheme of the flower morphology and its generative structures of ‘Honghu Hong’. (**b**) Longitudinal section at the stamen primordium differentiation stage, the arrows indicate the stamen primordium. (**c**) Longitudinal section at the pistil primordium differentiation stage, the arrows indicate the carpel. (**d**–**f**) Anther and anther appendagesat different developmental stages, the arrows indicate the anthers and the appendages respectivly. (**g**) Transverse section of the anther at the mature pollen stage. (**h**) Transverse section of the anther appendage at the young microspore stage. An, anther; Ap, appendage; Ca, carpel; E, epidermis; En, endothecium; F, filament; M, middle layer; MP, mature pollen; OB, osmiophilic body; P, parenchyma; Re, receptacle; St, stamen; T, tapetum; Te, Tepal; V, vascular elements. (**b**,**c**) Bar = 100 μm; (**d**,**e**) Bar = 0.3 mm; (**f**) Bar = 10 mm; (**g**,**h**) Bar = 10 μm.

**Figure 2 ijms-20-01015-f002:**
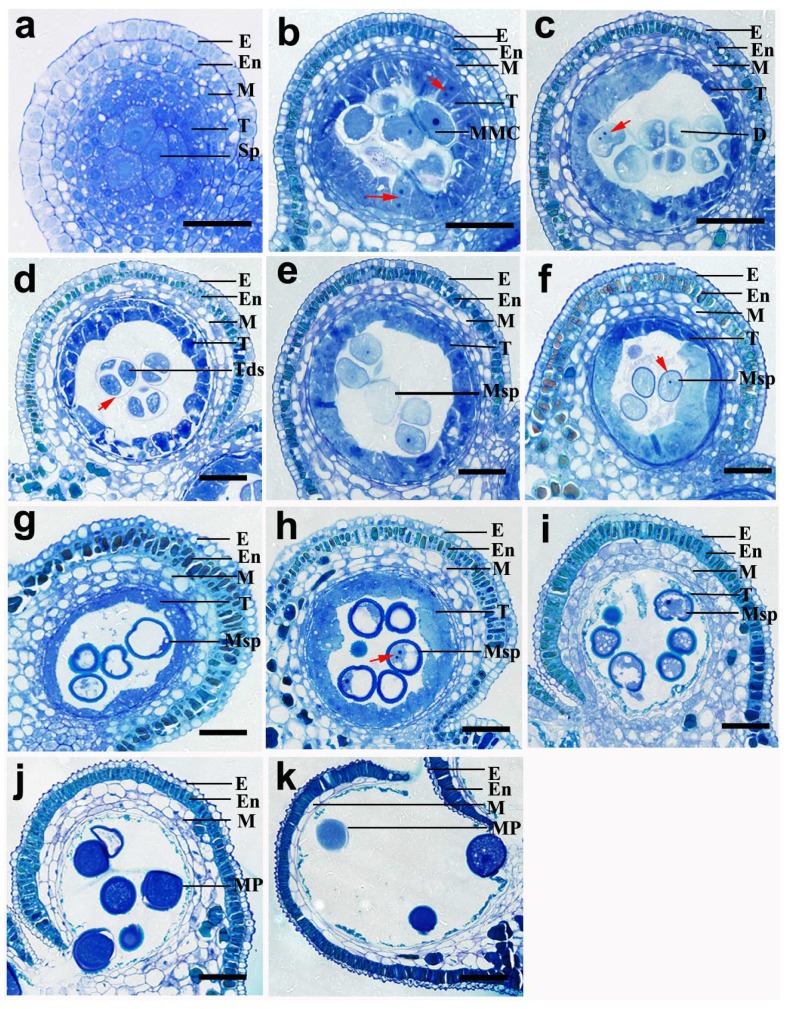
Cytological observation of locular anther development and pollen formation during the eleven developmental stages. (**a**) The sporogenous cell stage (stage 1). (**b**) The microspore mother cell stage (stage 2), the arrows indicate the mitosis process in the tapetal cells. (**c**) The dyads stage (stage 3), the arrow indicates the division process in dyads. (**d**) The tetrad stage (stage 4), the arrow indicates the callose wall around the tetrads. (**e**) The early young microspore stage (stage 5). (**f**) The middle young microspore stage (stage 6), the arrow indicates the primexine developed on the surface of the young microspore. (**g**) The late young microspore stage (stage 7). (**h**) The early bicellular pollen stage (stage 8), the arrow indicates the mitosis of nucleus in the young microspore. (**i**) The late bicellular pollen stage (stage 9). (**j**) The mature pollen stage (stage 10). (**k**) The anther dehiscence stage (stage 11). D, dyads; E, epidermis; En, endothecium; M, middle layer; MMC, microspore mother cell; MP, mature pollen; Msp, microspore; Sp, sporogenous cell; T, tapetum. Bar = 50 μm.

**Figure 3 ijms-20-01015-f003:**
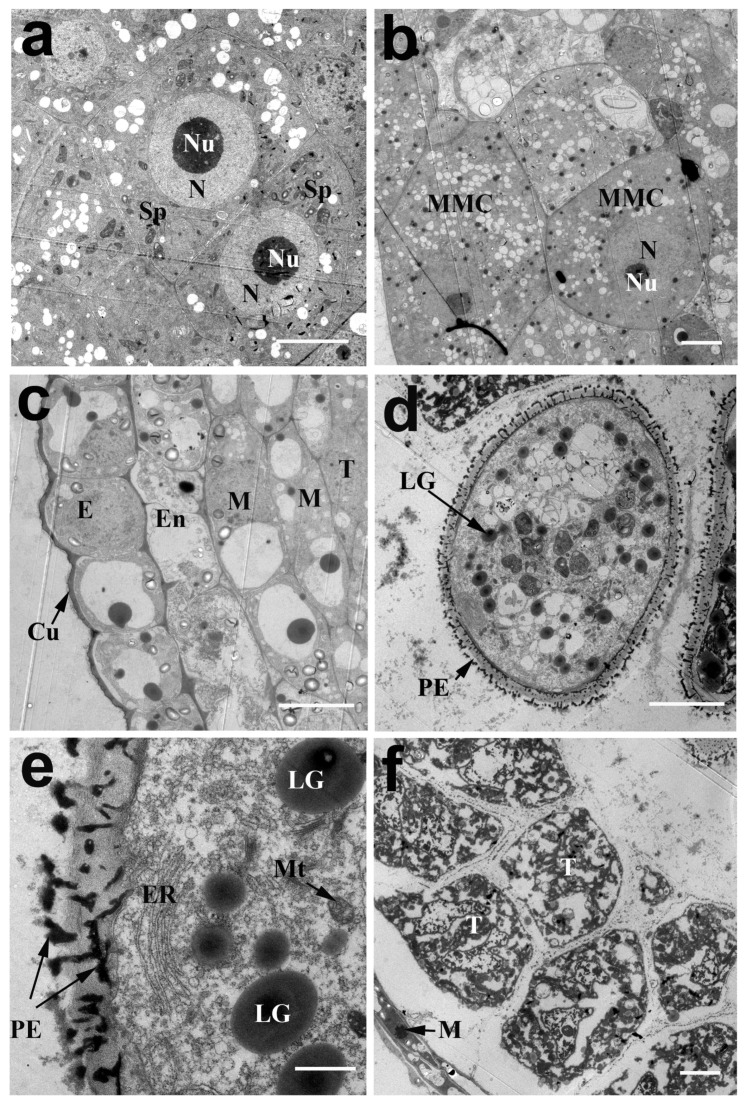
Transmission electron micrographs of cross-sections through anthers of *N. nucifera* ‘Honghu Hong’. (**a**) The sporogemous cells at the sporogenous cell stage. (**b**) The microspore mother cell at the MMC stage. (**c**) The anther wall comprising the epidermis, endothecium and middle layers at the MMC stage. The arrow indicates the epidermal cuticle. (**d**) The young microspores at the early young microspore stage, the arrows indicate the lipid granule and primexine respectively. (**e**) The magnification image of the young microspore wall at the early young microspore stage, the arrows indicate the primexine and mitochondria, respectively. (**f**) The magnification image of the tapetal cells at the early young microspore stage, the arrow indicates the middle layer. Cu, anther epidermal cuticle; E, epidermis; En, endothecium; ER, endoplasmic reticulum; LG, lipid granule; M, middle layer; MMC, microspore mother cell; Mt, mitochondria; N, nucleus; Nu, nucleolus; PE, primexine; SP, sporogenous cell; T, tapetum. (**a**–**d**,**f**) Bar = 5 μm; (**e**) Bar = 1 μm.

**Figure 4 ijms-20-01015-f004:**
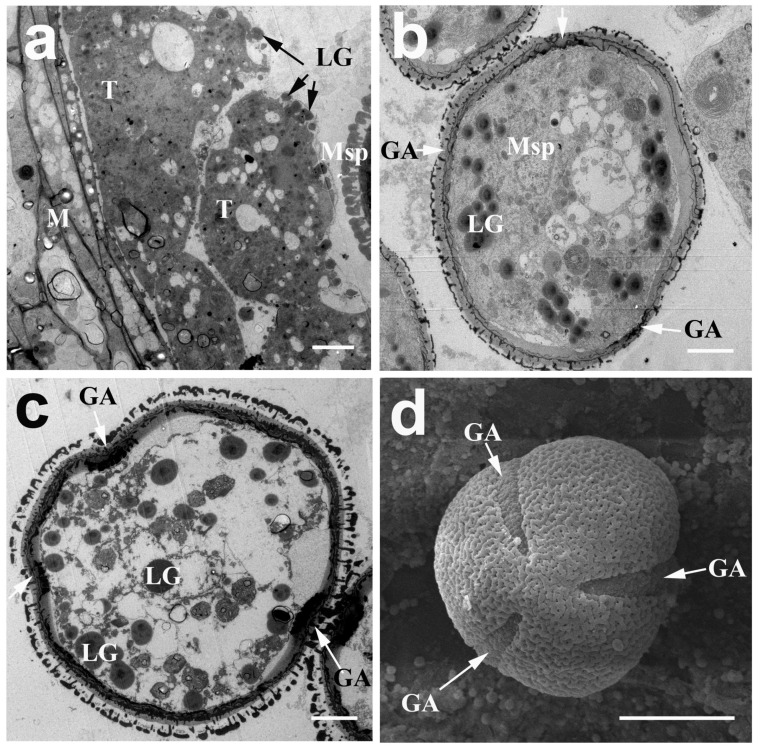
Transmission electron micrographs (**a**–**c**) and scanning electron micrographs (**d**) of the microspore and tapetum of *N. nucifera* ‘Honghu Hong’. (**a**) Numerous lipid granules were secreted from tapetum to the locule at the late young microspore stage. (**b**) Microspore at the middle young microspore stage. (**c**) Microspore at the late young microspore stage. (**d**) Microspore at the mature pollen stage. The arrows indicate the germination apertures in (**b**–**d**). GA, germination aperture; LG, lipid granule; M, middle layer; Msp, microspore; T, tapetum. (**a**–**c**) Bar = 5 μm; (**d**) Bar = 30 μm.

**Figure 5 ijms-20-01015-f005:**
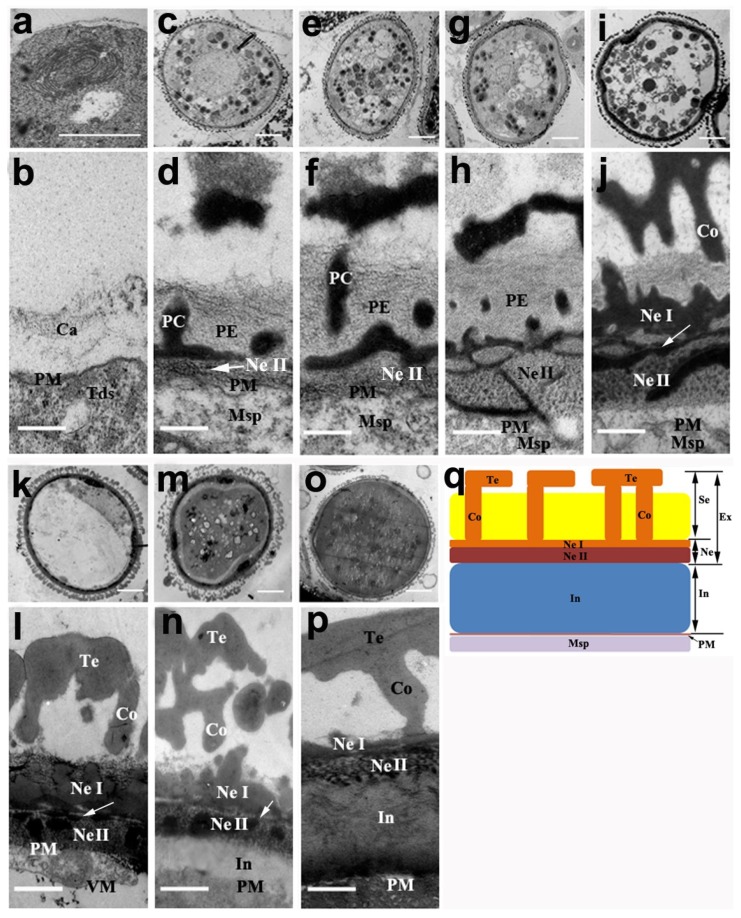
The development of the pollen wall in the transmission electron microscopy (**a**–**p**) of *N. nucifera* ‘Honghu Hong’. (**a**) Microspore at the tetrad stage. (**b**) The callose wall was the first of several layers deposited on the microspore surface at the tetrad stage. (**c**) Microspore at the early young microspore stage. (**d**) Detail of the microspore wall at the early young microspore stage, the arrow indicates the nexine II. (**e**) Microspore at the early young microspore stage, slightly later than Figure 5d. (**f**) Detail of the microspore wall at the early young microspore stage, slightly later than Figure 5d. (**g**) Microspore at the middle young microspore stage. (**h**) Enlargement of the microspore wall at the middle young microspore stage. (**i**) Microspore at the late young microspore stage. (**j**) Detail of the microspore wall at the late young microspore stage. (**k**) Microspore at the early bicellular pollen stage. (**l**) Detail of the microspore wall at the early bicellular pollen stage. (**m**) Microspore at the late bicellular pollen stage. (**n**) Detail of the pollen wall at the late bicellular pollen stage. (**o**) Microspore at the mature pollen stage. (**p**) Detail of the pollen wall at the mature pollen stage. (**q**) Proposed model of the pollen exine structure. The white arrow indicates the formation of white-line-center between nexine I and nexine II in (**j**,**l**,**n**). Ca, callose; Co, columella; In, intine; Msp, microspore; Ne, nexine; Ne I, nexine I; Ne II, nexine II; PC, precolumella; PE, primexine; PM, plasma membrane; PN, prenexine; Se, sexine; Tds, tetrads; Te, tectum; VM, vacuole membrane. (**a**,**c**,**e**,**g**,**I**,**k**,**m**,**o**) Bar = 5 μm; (**b**,**d**,**f**,**h**,**j**) Bar = 0.5 μm; (**l**,**n**,**p**) Bar = 1.0 μm.

**Figure 6 ijms-20-01015-f006:**
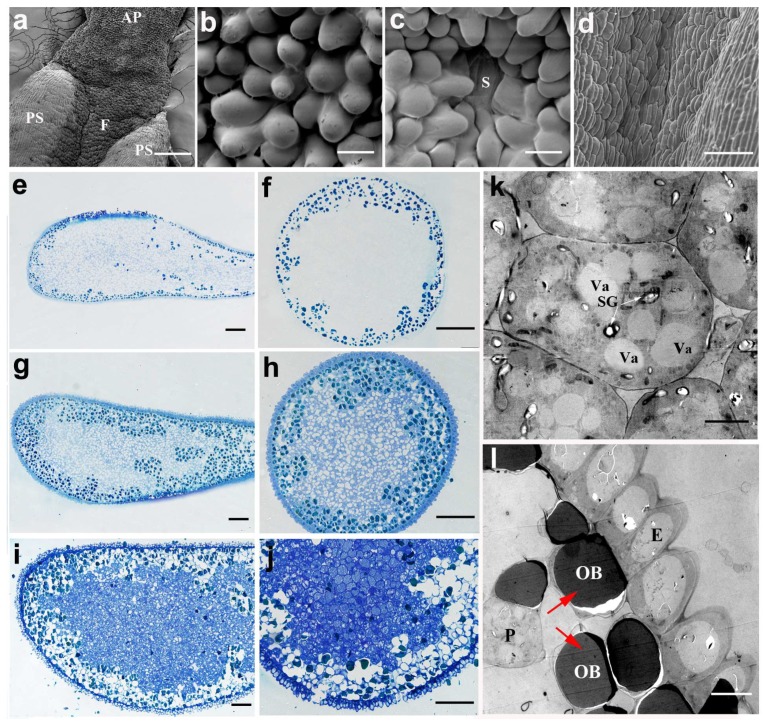
The structure and development of the anther appendage of *N. nucifera* ‘Honghu Hong’. (**a**) View of the appendage–anther junction. (**b**) The surface structure of the appendage at the mature pollen stage. (**c**) The surface structure of the filament at the mature pollen stage. (**d**) The surface structure of the anther at the mature pollen stage. (**e**–**j**) Longitudinal and lateral sections of the intact appendage at the MMC stage (**e**,**f**), the young microspore stage (**g**,**h**) and the mature stage (**i**,**j**). (**k**) The transmission electron micrograph of a parenchyma cell, the white arrows indicate the starch granules. (**l**) The transmission electron micrograph of the epidermis, the red arrows indicate the osmiophilic body in the parenchyma cells. AP, appendage; E, epidermis; F, filament; OB, osmiophilic body; P, parenchyma; PS, pollen sac; S, stomate; SG, starch granule; Va, vacuole. (**a**) Bar = 300 μm; (**b**,**c**) Bar = 10 μm; (**d**) Bar = 50 μm; (**e**–**j**) Bar = 100 μm; (**k**) Bar = 5 μm; (**l**) Bar = 10 μm.

**Figure 7 ijms-20-01015-f007:**
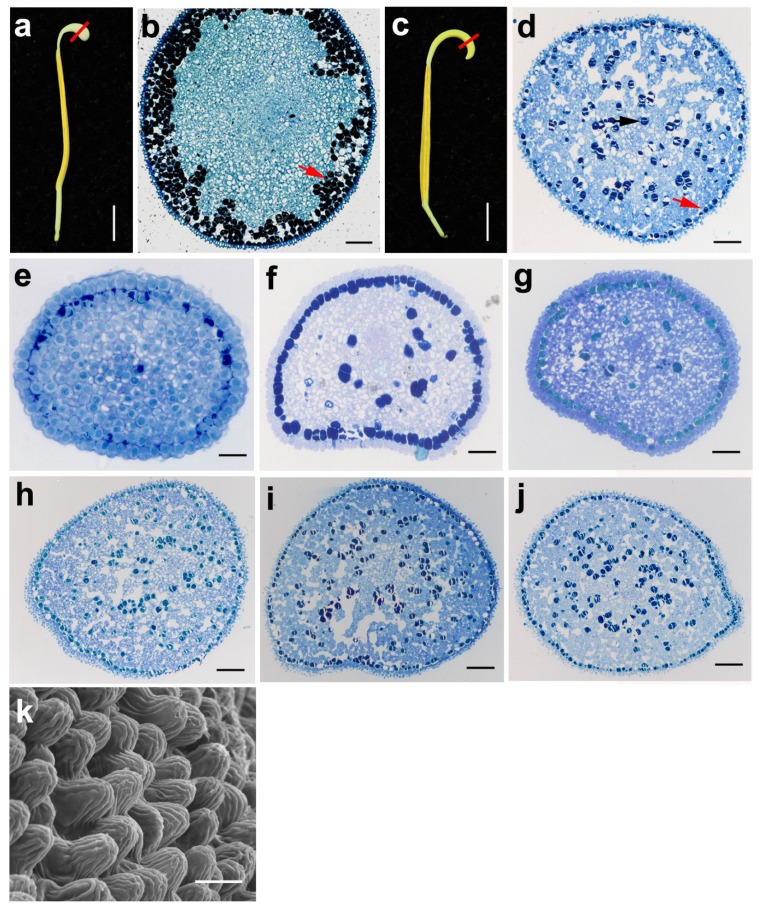
The structural comparison of the anther and its appendage between *N. nucifera* (**a**,**b**) and *N. lutea* (**c**–**k**). (**a**) The morphology of the mature anther; the red line indicates the place for cross section. (**b**) Cross section of the anther appendage, the red arrow indicates the osmiophilic bodies accumulated with the multiple layers under the epidermis cells. (**c**) The morphology of the mature anther; the red line indicates the place for the cross section. (**d**) Cross section of the appendage, the arrows indicate the osmiophilic bodies randomly scattered in the parenchyma cells. (**e**–**j**) Transverse section of the anther appendage at the microspore mother cell stage (**e**), the tetrad stage (**f**), the early young microspore stage (**g**), the middle free spore stage (**h**), the late young microspore stage (**i**) and the mature stage (**j**). (**k**) The scanning electron micrograph of the surface structure of the anther appendage at the mature stage. (**a**,**c**) Bar = 1 cm; (**b**,**d**,**g**–**j**) Bar = 100 μm; (**e**,**f**) Bar = 20 μm; (**k**) Bar = 10 μm.

**Table 1 ijms-20-01015-t001:** Detailed description of the anther and pollen developmental stages of *N**. nucifera* ‘Honghu Hong’.

Stage	Bud Length (cm)	Anther Length (mm)	Appendage Length (mm)	Pollen Development Stage	Significant Events
1	<1.5	<1.5	<0.5	Sporogenous cell	Formation of four-layered anther wall and pre-meiosis DNA synthesis
2	1.5–2.0	<1.5	0.5	MMC	Formation of microspore mother cells and tapetum layer. Mitosis of tapetal cells
3	2.0–2.8	2.0–5.0	<1.0	Dyads	Meiosis I
4	2.8–4.0	5.0–6.0	1.0–1.5	Tetrad	Formation of four haploid spores
5	4.0–5.0	6.0–11.0	1.5–3.0	Early young microspore	Degradation of callose wall. Formation of uninucleated gametophyte and primexine
6	5.0–6.0	11.0–15.0	3.0–4.5	Middle young microspore	Formation of exine
7	6.0–6.5	15.0–15.5	4.5–5.0	Late young microspore	Formation of large central vacuoles
8	6.5–7.0	15.5–16.0	5.0	Early bicellular pollen	Formation of vegetative nucleus and a generative nucleus by pollen mitosis I
9	7.0–9.0	16.0	5.0	Late bicellular pollen	Formation of intine
10	9.0–10.0	16.0–16.5	5.0	Mature pollen	Accumulation of starch grains
11	10.0 (fully open)	16.6	5.0	Anther dehiscence	Anther dehiscence and pollen released

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
