# Peer review of "Histological and Cytological Characterization of Anther and Appendage Development in Asian Lotus (Nelumbo nucifera Gaertn.)"

_ijms, 2019, doi:10.3390/ijms20051015_

Round 1

Reviewer 1 Report

Dear authors,

The experimental design for your research was logical and coherent, and the presented results are well explained and with good detail. Regarding the content of the article, it has novelty as no other publication has ever described developmental stages in this species. The manuscript is well written in general but will need to be reviewed by an English native speaker as there are some sentences that are grammatically incorrect and make it difficult to understand what the authors are trying to explain. Overall is a good article with interesting results, however they don’t seem to fit under the scope of the IJMS as they do not present any molecular methodology or data. 

Author Response

The experimental design for your research was logical and coherent, and the presented results are well explained and with good detail. Regarding the content of the article, it has novelty as no other publication has ever described developmental stages in this species. The manuscript is well written in general but will need to be reviewed by an English native speaker as there are some sentences that are grammatically incorrect and make it difficult to understand what the authors are trying to explain. Overall is a good article with interesting results, however they don’t seem to fit under the scope of the IJMS as they do not present any molecular methodology or data. 

Response: Thank you very much for your positive comments. The grammar and language of the manuscript was revised by MDPI English Editing Service.

Reviewer 2 Report

There are a number of grammatical, syntax and language errors in the in manuscript in particular in Paragraph Lines 63 to 75. The content of the manuscript is fine and if the language can be improved it should be published.,

Author Response

There are a number of grammatical, syntax and language errors in the in manuscript in particular in Paragraph Lines 63 to 75. The content of the manuscript is fine and if the language can be improved it should be published.,

Response: Thank you very much for the positive comments. The MDPI English Editing Service has revised the manuscript thoroughly and corrected the grammar errors.

We revised the sentence in line 63-74 ‘The pollen development of the American lotus has been described and can be divided into five different stages [9]. However, the development of the pollen and anther of Asian lotus, a more economically important species, still remains unknown. Due to a long history of geographical isolation and evolution, large differences exist in the morphology between these two species, especially in the stamen, anther pigment and appendage shape. Although both the American lotus and Asian lotus can easily hybridize each other to produce fertile seeds, the reproductive ability of their hybrid offspring has largely declined [4, 10]. To better understand the process and characteristics of the male reproductive development of Asian lotus in comparison with previous reports on the American lotus, we investigated the flower and anther structure, anther morphogenesis, and pollen formation of a wild type Asian lotus. Exploring the developmental events in the lotus plant, particularly into its reproductive mechanism, will improve our knowledge on species identity and differentiation.’

Reviewer 3 Report

. The paper describes anther and pollen development on histological and scanning and electron microscopy levels. Lack of molecular methods or even discussion on molecular basis of floral organs development.

Lack of novelty, anther and pollen structure and ultrastructure and development were described in other species of Nelumbo and Authors should not expect that in other species of the genus these processes are different. The anther characteristics (cell wall structure), microsporogenesis and microgametogenesis are well described processes in Angiosperms.

Comments and suggestions are in attached pdf of the paper.

Detailed comments:

The aims of the study are not clearly presented.  Why Authors select ‘Honghu Hong’ cultivar of Nelumbo nucifera for the study but not plant from nature?

In Results figures are not quoted in order in the text

Some parts of Results are in fact Discussion with literature citations.

Regarding that anther and pollen structure and development are well described in model plants and also in other species, there is no reason to describe these processes in details in this paper. The paper is too wordy.

General comment to Discussion: in most parts the results are repeated in Discussion even figures and tables are cited.

In Materials and Methods there is lack of information about number of samples used (e.g. number of flowers)

Lack of conclusion; in Conclusion paragraph there is summary but not conclusion.

Some sentences are not clear, the language has to be corrected by style and grammar.

Some conclusions are not based on detailed analysis.

Figure captions need corrections according suggestions in pdf of the manuscript

Fig. 1. a – Is this scheme made by the authors or is taken from literature. If so the references has to be added; d - the stages of floral bud development are not informative could be removed.

Fig. 2. Some micrographs are not good quality and the details are not recognizable. Comments are in the pdf of the manuscript.

Author Response

The aims of the study are not clearly presented.  Why Authors select ‘Honghu Hong’ cultivar of Nelumbo nucifera for the study but not plant from nature?

Response:

Thank you for your insightful comments. We rewrote the aim of the study.

The aim of the study is to describe the cytological process of microsporogenesis and determine the developmental characteristics of anther and pollen in Asian lotus. Due to a long history of geographical isolation and evolution, large differences showed in morphology, especially in stamen, anther pigment and appendage shape between Asian lotus and American lotus. These two species can hybridize each other to produce a few fertile seeds, and the reproductive ability of their hybrid offspring has largely declined. So it is necessary to analyze the difference of pollen development between these two species. In this manuscript, the developmental differentiation of pollen, pollen wall and appendage were identified and a standard stage classification of the pollen development process was presented. Our research is helpful to provide a reference of the pollen development process of lotus for further study at a molecular level. These cytological data will support further genetic hybridization among two species of lotus.

We revised the aim of this study as follow (details see the revised manuscript line 69-74):

To better understand the process and characteristics of the male reproductive development of Asian lotus in comparison with previous reports on the American lotus, we investigated the flower and anther structure, anther morphogenesis, and pollen formation of a wild type Asian lotus. Exploring the developmental events in the lotus plant, particularly into its reproductive mechanism, will improve our knowledge on species identity and differentiation.

(2) ‘Honghu Hong’ cultivar of N. nucifera is actually selected from a natural wild population of Asian lotus in China. In order to avoid ambiguity, we deleted the ‘cultivar of ’ in the revised manuscript (line 72). Also, we give more information of this Asian lotus in ‘Materials and Methods’ part (line 331-332).

N. nucifera ‘Honghu Hong’ (large plant, red and single flower, a wild type introduced from the natural population of Hong Lake in Hubei Province, China) and American yellow lotus (large plant, light-yellow and single flower, wild type from Miami, Florida, USA) were used as experimental materials.”

In Results figures are not quoted in order in the text

Response: We deleted the disordered figure number “(Figure 7a and 7c)” in line 92-93. We also deleted the unnecessary photograph d in Figure 1 and reordered the Figure numbers in the revised manuscript (line 77 to line 96).

Some parts of Results are in fact Discussion with literature citations.

Response: We deleted these unrelated results from Results part in the revised manuscript. For example,

(1) In line 77, we deleted sentence of “In higher plants, male reproductive development is a complex biological process that includes stamen identity from the floral meristem, anther morphogenesis and pollen formation within the flower [11].”

(2) In line 109, we deleted “In a model plant, such as Arabidopsis (the model plant for eudicots) and rice (the model plant for monocots), the structure of anther is consist of four somatic cell layers: epidermis, endothecium, middle layer, tapetum and the inner microsporocytes [13]. Anther development has been divided into 14 stages based on morphological features [14, 15].”

In line 169, we deleted “These evidences suggested that species-specific pollen wall patterning was inherited by the microspores [16].”

(3) In line 187, we deleted sentences “The pollen wall is a multilayered structure made up of a pectocellulosic intine surrounded by a sporopollenin-based exine, which contains two layers: the inner nexine (including nexine I and nexine II) and the outer sexine (including tectum and columlla) [16]. To precisely describe subcellular changes of pollen wall formation, we observed the pollens by TEM method (Figure 5).”

Regarding that anther and pollen structure and development are well described in model plants and also in other species, there is no reason to describe these processes in details in this paper. The paper is too wordy.

Response: Thank you for the useful comments. We deleted the repeat and unnecessary description and revised this part thoroughly. Details please see ‘2.2. The Development of Pollen in N. nucifera’ part in the revised paper.

(1) In line 135, we deleted sentences ‘The structure of epidermis, endothecium, middle layer and tapetum can be seen clearly (Figure 2a). The middle layer with multiple layers in lotus anther is a unique characteristic different from the model plants such as Arabidopsis and rice.’

(2) In line 154, we deleted “The most important character of these two stages is that the entire dyads and tetrads were surrounded with callose (Figure 2d, red arrow indicated). The tapetal cells at these two stages also closely abut and are deeply stained.”

(3) In line 158-159, we deleted “The tapetum begins to dissociate at this stage with its disintegration and degradation (Figure 3f).” and “The significant characteristic of the young microspores stage was numerous mitochondria and lipid granule filled in the cytoplasm.”

General comment to Discussion: in most parts the results are repeated in Discussion even figures and tables are cited.

Response: Thank you for the insightful comments. We deleted the repeated sentences in Discussion part. 

(1) In line 267, we deleted the whole paragraph of “Lotus is a very important ornamental, aquatic vegetable and medicinal plant in the world. The American lotus has little diversity at population level and a very few cultivars, while Asian lotus is richer in germplasm and has over 2000 cultivars through a long history of artificial hybridization and selection [4, 17]. The pollen and anther development of American lotus have been explored nearly twenty years before [6]. However, it remains largely unknown on Asian lotus before our study. This is the first report to comprehensively describe pollen and anther development in Asian lotus. The major events of morphogenesis and specialized cytological ontogeny are summarized and discussed below.”

(2) In line 274, we deleted “It secretes large amounts of carbohydrates and lipids into locule to provide nutrition for the microspore development. With the degradation of tapetum at the young microspore stage, the numerous lipids particles (orbicular tapetosomes) were secreted from the tapetum to the surface of microspore to form the sporopollenin (Figure 4a).”

(3) In line 307, we deleted “At the early stages of anther development, the length of anther and appendage in Asian lotus gradually elongated (from 10 mm to 15 mm) while the length of flower buds was almost the same (about 5 cm). Until the late young microspore stage, the flower buds and filaments continued to grow, but the length of anther and appendage almost no longer extended (Table 1).”

In Materials and Methods there is lack of information about number of samples used (e.g. number of flowers)

Response: We added the information on sample in line 331-337 in “Materials and Methods”.

N. nucifera ‘Honghu Hong’ (large plant, red and single flower, a wild type introduced from the natural population of Hong Lake in Hubei Province, China) and American yellow lotus (large plant, light-yellow and single flower, wild type from Miami, Florida, USA) were used as experimental materials. Both lotuses were cultivated in the International Nelumbo Collection (INC) located at Shanghai Chenshan Botanical Garden, Shanghai, China. Three floral buds in each lotus were collected at different stages and thereafter ten fresh anthers in each bud were isolated for fixing in FAA solution.

Lack of conclusion; in Conclusion paragraph there is summary but not conclusion.

Response: Thank you for your useful advice, we revised the conclusion thoroughly. Details see in line 367-385 of the manuscript.

The characteristic development of the anther and appendage and cytological process of microsporogenesis of the Asian lotus were investigated.  Eleven developmental stages were proposed for the pollens according to the biological events. Through continuous cytological observations and analysis, the key biological cytological events of male reproductive development at different stages of the Asian lotus were determined. This is helpful for providing a reference, or even a standard stage classification of the pollen development process of lotus for further study at a molecular level. The development and differentiation of pollen and pollen wall were the basis of understanding the male reproductive development process of lotus. Therefore, these cytological data will support further genetic hybridization among two species of lotus. Moreover, the comparative analysis of the anther and microspore formation process will provide a positive contribution to the evolution between terrestrial and aquatic plants. The anther’s appendage is a derivative structure of filaments and it forms before the pollen sac. Furthermore, the appendage morphology and the content of osmiophilic bodies in the appendage were obviously different between the Asian lotus and American lotus. It is unclear whether this component is an evolutionary process or a degenerate residue in lotus. Determination of the biological function of anther appendage is an interesting and challenging topic for future study. In short, this is the first report to comprehensively describe the pollen and anther development in Asian lotus. The results expanded our knowledge on the pollen and appendage development of Nelumbo and will also improve understanding of the identity and differentiation of the two lotus species.

Some sentences are not clear, the language has to be corrected by style and grammar.

Response: Thanks, the style and grammar were corrected by the MDPI English Editing Service. We also deleted the vague sentences or answer the questions in the PDF manuscript, and revised the manuscript carefully. Please see details in the manuscript. For example,

(1) In line 321-323, we revised a sentence as “Besides this, it has been proposed to be the source of fragrance in lotus (for example, in Vietnam the anther appendages are mixed with tea to increase fragrance).”

(2) In line 327-328, we deleted “This structural difference may cause the appendages easier to loss water in American lotus than Asian lotus (Figure 7k and 7l).”

Some conclusions are not based on detailed analysis.

Response: Thanks, we deleted some inappropriate sentences. For example,

In line 304, we deleted sentences of “Although American lotus and Asian lotus are apart far away in geographical position and taken as two species, both can hybridize each other and produce fertile offspring [7]. The common characteristics of pollen structures and developmental process between these two species supported their closely genetic relationship and a high feasibility of interspecific hybridization.”

In line 327, we deleted sentence of “This structural difference may cause the appendages easier to loss water in American lotus than Asian lotus (Figure 7k and 7l).”

Figure captions need corrections according suggestions in pdf of the manuscript

Fig. 1. a – Is this scheme made by the authors or is taken from literature. If so the references has to be added; d - the stages of floral bud development are not informative could be removed.

Response: Thanks. This scheme was made by authors and did not cite from any other reference.

We deleted the photograph d from the Figure 1 and reordered the Figure numbers in Figure 1. We also added the abbreviation of appendage (Ap), anther (An) and filament (F) in Figure 1d, 1e and 1f. Accordingly, we revised the manuscript from line 81 to line 107.

Figure f and Figure e are not the same stages. Figure f is a later stage of anther, and it is not a magnification of e. In Figure e, anthers and appendages were almost indistinguishable under the microscope, In Figure f, anthers and appendages can be obviously observed with the Naked Eye. The pollen sac was firstly appeared around the filaments at this stage.

Fig. 2. Some micrographs are not good quality and the details are not recognizable. Comments are in the pdf of the manuscript.

Response: We improved the micrographs quality of Figure 2 and other Figures. In Figure 2, we changed the photograph a, c and e. In figure 2c, red arrow indicated dividing dyads. Details responses are list in the PDF manuscript and revised manuscript.

Reviewer 4 Report

Line 2. In the title, I suggest including the word "Histology". So .."Histological and Cytological characterization of anther ..."..is fine.

Line 38. Please could be put a few references about medicinal, edible and cultural things?

Line 118-119. Include N. nucifera (Honghu Hong) 

Line 133. Same comment of line118.

Line 283-284. Please indicate TEM (transmission electron micrograph). The surface appendage in N. nucifera is smooth versus linearly rough in N. lutea. Probably this aspect could be inserted in general discussion or conclusions (lines 402-403)

In the photographs of the appendage on page 11 (Figures a and c) it is obvious that there are macro differences between species.

Author Response

Comments and Suggestions for Authors

Line 2. In the title, I suggest including the word "Histology". So .."Histological and Cytological characterization of anther ..."..is fine.

Response: Thank you for your useful comments. We changed the title into “Histological and Cytological characterization of anther and appendage development in Asian lotus (Nelumbo nucifera Gaertn.)”

Line 38. Please could be put a few references about medicinal, edible and cultural things?

Response: Thanks. We added three review references about medicinal, edible and cultural things, please see the References part.

1. Zhu M. Z., Liu T. and Guo M. Q. Current advances in the metabolomics study on lotus seeds. Front. Plant Sci. 2016, 7, 891. doi: 10.3389/fpls.2016.00891.

2. Sharma B. R., Gautam L. N. S., Adhikari D. and Karki R. A comprehensive review on chemical profiling of Nelumbo Nucifera: potential for drug development. Phytother. Res. 2016, doi: 10.1002/ptr.5732. 

3. Guo, H.B. Cultivation of lotus (Nelumbo nucifera Gaertn.ssp nucifera) and its utilization in China. Genet. Resour. Crop Evol. 2009, 56, 323–330. doi: 10.1007/s10722-008-9366-2.

 Line 118-119. Include N. nucifera (Honghu Hong) 

Response: Thanks. According to other reviewer’s suggestion, we deleted this sentence in line 118-119 “To better describe the characteristics of pollen development in lotus, we analyzed the cellular changes occurring in the anthers of ‘Honghu Hong’ by light microscopy and TEM observation of transverse sections,……..”. Please see line 109 in the revised manuscript.

Line 133. Same comment of line118.

Response: Thanks. We added the N. nucifera before ‘Honghu Hong’. Please see the title of Table 1 in line 146.

Line 283-284. Please indicate TEM (transmission electron micrograph). The surface appendage in N. nucifera is smooth versus linearly rough in N. lutea. Probably this aspect could be inserted in general discussion or conclusions (lines 402-403)

Response: Thank you for your useful comments. It was scanning electron micrograph (SEM) in Figure 7k and 7l. According to other reviewer’s comment, we deleted Figure 7k and revised Figure 7l with legend as “The scanning electron micrograph of surface structure of anther appendage at the mature stage”. Details please see in Figure 7 in line 263-264.

We described the difference of appendage surface between N. nucifera and N. lutea in line 241-243 (Results part) and line 377-382 (Conclusions part).

In the photographs of the appendage on page 11 (Figures a and c) it is obvious that there are macro differences between species.

Response: Yes. We described these differences in the Results part. In line 91-93 “The appendage of the Asian lotus stamen was long, oval shaped, and lay at the top of the connective (Figure 1f). It is usually milky white or sometimes pink to rosy red in color for Asian lotus cultivars, while the appendage of the American lotus is bright yellow with a sickle-like shape.”

Round 2

Reviewer 1 Report

I believe that the content of the manuscript although is well described and presented, does not fit the scope of the International Journal of Molecular Sciences. Authors only made grammar corrections to improve the text. 

Author Response

Comments and Suggestions for Authors

I believe that the content of the manuscript although is well described and presented, does not fit the scope of the International Journal of Molecular Sciences. Authors only made grammar corrections to improve the text. 

Response: Thank you very much for your positive comments about the manuscript. As described in the IJMS, “….an international peer-reviewed open access journal providing an advanced forum for biochemistry, molecular and cell biology, and molecular biophysics.” (https://www.mdpi.com/journal/ijms). Our research should fit the field of cell biology, although the majority of previously published papers in this journal are related to molecular biology.

Due to the lagging research on reproductive development and the difficulty in establishing transformation system in Nelumbo, the molecular biology studies related to lotus have been rarely reported compared with the model plants (such as Arabidopsis and rice). The goal of this study is mainly focused on understanding the characteristics of anther and pollen development of lotus by means of cell biology. Currently, we are focusing on determining the lipid components of appendages in Asian lotus and American lotus. We also work on establishment of lotus’s transformation system and will answer more deep questions related to development of some important traits in lotus by the molecular approaches.

Reviewer 3 Report

The manuscript was revised according to suggestions and Authors responded for comments.

Still manuscript need correction.

There is no need to cite figures and tables in Discussion. These citations should be minimalized to most important for understanding

Conclusions are not acceptable.

Reviewer's comments are included into the pdf of revised version

Reviewer’s suggestion:

Conclusion

This is the first report of the pollen and anther development in Asian lotus. The results expanded our knowledge on the pollen and anther appendage development of Nelumbo and will also improve understanding of the identity and differentiation of the two lotus species.

The key cytological events of eleven stages of anther and pollen development of the Asian lotus documented in details in this study are good base to analyze the differences between lotus species, selecting taxonomically important traits.

 Description of regular anther and pollen development of Asian lotus allows to recognize abnormalities at different stages which could lead to male sterility. 

Cytological data of anther and pollen development are useful in further crossbreeding experiments used for the commercial production of hybrids between two lotus species.

The comparative analysis of the anther and microspore formation process will provide a important contribution to the evolution of terrestrial and aquatic plants.

The anther’s appendage traits discriminate Asian and American lotuses. Determination of the biological function of anther appendage is an interesting and challenging topic for future study.

Author Response

Comments and Suggestions for Authors

The manuscript was revised according to suggestions and Authors responded for comments.

Still manuscript need correction.

There is no need to cite figures and tables in Discussion. These citations should be minimalized to most important for understanding

Response: Thanks, we deleted the citations of figures and tables in Discussion part. The details can be seen in the manuscript. For examples:

Line 275, ‘(Figure 4a)’ was deleted from Discussion.

Line 284, ‘(Figure 2)’ was deleted.

Line 288, ‘(Figure 2j,k)’ were deleted.

Line 306, ‘(Figure 5j, 5l and 5n)’ were deleted.

Line 313, ‘(Table 1)’ was deleted.

Line 316, ‘(Figure 1f)’ was deleted.

Line 326, ‘(Figure 7b, as indicated by the red arrow),’ was deleted.

Line 327, ‘(Figure 7d, as indicated by the black arrow)’ was deleted.

Line 329, ’(Figure 7e, j)’ was deleted.

Conclusions are not acceptable.

Response: Thank you very much for your constructive comments, we revised the conclusion as you suggested. Also, we responded all the comments point by point in the revised PDF file. Please see details in the PDF file.